# Proteomic Analysis of a Rat Streptozotocin Model Shows Dysregulated Biological Pathways Implicated in Alzheimer’s Disease

**DOI:** 10.3390/ijms25052772

**Published:** 2024-02-28

**Authors:** Esdras Matheus Gomes da Silva, Juliana S. G. Fischer, Isadora de Lourdes Signorini Souza, Amanda Caroline Camillo Andrade, Leonardo de Castro e Souza, Marcos Kaoann de Andrade, Paulo C. Carvalho, Ricardo Lehtonen Rodrigues Souza, Maria Aparecida Barbato Frazao Vital, Fabio Passetti

**Affiliations:** 1Instituto Carlos Chagas, FIOCRUZ, Curitiba 81310-020, PR, Brazil; 2Laboratory of Toxinology, Oswaldo Cruz Institute, Fiocruz, Rio de Janeiro 21040-361, RJ, Brazil; 3Laboratory of Polymorphisms and Linkage, Federal University of Paraná, Curitiba 81530-001, PR, Brazil; 4Department of Pharmacology, Federal University of Paraná, Curitiba 80050-540, PR, Brazil

**Keywords:** mass spectrometry, qRT-PCR, pre-frontal cortex, hippocampus, mitochondrial membrane, ribosome, insulin secretion, chaperone complex

## Abstract

Alzheimer’s Disease (AD) is an age-related neurodegenerative disorder characterized by progressive memory loss and cognitive impairment, affecting 35 million individuals worldwide. Intracerebroventricular (ICV) injection of low to moderate doses of streptozotocin (STZ) in adult male Wistar rats can reproduce classical physiopathological hallmarks of AD. This biological model is known as ICV-STZ. Most studies are focused on the description of behavioral and morphological aspects of the ICV-STZ model. However, knowledge regarding the molecular aspects of the ICV-STZ model is still incipient. Therefore, this work is a first attempt to provide a wide proteome description of the ICV-STZ model based on mass spectrometry (MS). To achieve that, samples from the pre-frontal cortex (PFC) and hippocampus (HPC) of the ICV-STZ model and control (wild-type) were used. Differential protein abundance, pathway, and network analysis were performed based on the protein identification and quantification of the samples. Our analysis revealed dysregulated biological pathways implicated in the early stages of late-onset Alzheimer’s disease (LOAD), based on differentially abundant proteins (DAPs). Some of these DAPs had their mRNA expression further investigated through qRT-PCR. Our results shed light on the AD onset and demonstrate the ICV-STZ as a valid model for LOAD proteome description.

## 1. Introduction

Alzheimer’s disease (AD) is an age-related neurodegenerative disorder characterized by progressive memory loss and cognitive impairment [1]. Currently, AD accounts for nearly 80% of dementia cases, affecting 35 million individuals worldwide [2]. The pre-frontal cortex (PFC) and hippocampus (HPC) are the brain structures particularly affected by this disease [3] since they are critically important in the brain’s memory system [4]. Late-onset Alzheimer’s disease (LOAD) is the most prevalent form of the disease [5]. Although the etiology of LOAD is still unclear, extracellular amyloid β (Aβ) peptides and intracellular hyperphosphorylated tau (p-tau) accumulation are classical molecular disturbances observed in the patient’s brain [6,7].

The impairment of cerebral glucose metabolism was observed in postmortem LOAD patients’ brains [8,9]. These findings were further consolidated leading to the hypothesis that a signal transduction failure of the cerebral insulin receptor could trigger the onset of the disease [10]. Streptozotocin (STZ) is a diabetogenic compound [11] generally used to produce experimental non-transgenic animal models of diabetes [12]. Studies conducted by Sigfried Hoyer in the early 1990s demonstrated that an intracerebroventricular (ICV) injection of low to moderate doses of STZ in adult male Wistar rats could cause an insulin resistant state in the brain and reproduce the classical physiopathological hallmarks of LOAD [13,14]. The ICV-STZ animal model has been considered as an appropriate model for the study of LOAD. Indeed, the ICV-STZ injection into rats induces progressive deficits in learning and memory [15,16] and neurochemical and neuropathological alterations in the brain that resemble those found in LOAD patients [17,18]. Since then, the ICV-STZ model has been widely applied to test the insulin resistance hypothesis of LOAD [19].

Most studies using the ICV-STZ model are focused on the description of the behavioral [20] and morphological [21] aspects of the disease. The molecular factors of the ICV-STZ model are often assessed through techniques that target only a few genes, such as qRT-PCR [22] and Western blot [23]. Mass spectrometry (MS)-based proteomics is a powerful analytical tool commonly used to study the proteome profile of LOAD’s brain [24]. Gaining insight into the molecular intricacies of the ICV-STZ model through high-throughput methodologies could offer a more comprehensive understanding of its application in the field of dementia. Therefore, this work is a first attempt to provide a wide proteome description based on MS proteomics of the ICV-STZ model’s PFC and HPC. Our analysis revealed dysregulated biological pathways implicated in the early stages of LOAD, based on differentially abundant proteins (DAPs). A set of the DAPs were selected to have their mRNA expression further investigated through quantitative Real-Time Polymerase Chain Reaction (qRT-PCR). Our results could be used as a baseline for future studies in the field.

## 2. Results

### 2.1. Overview of the ICV-STZ Model’s Proteome

Aiming for a qualitative analysis, a database search was performed, combining all biological replicates of each dataset (PFC and HPC) according to their groups (STZ and control). As a result, the following protein sequence numbers were identified: (1) PFC dataset: 6992 proteins from the STZ group (42% with at least one unique peptide, i.e., existing only in one protein) and 6629 proteins from the control group (42% unique); (2) HPC dataset: 6693 proteins from the STZ group (42% unique) and 6269 proteins from the control group (41% unique) (Table 1). As depicted in Figure 1, most of the identified proteins (Figure 1A) and peptides (Figure 1B) were common to both groups (STZ and control) in both datasets (PFC and HPC). All the proteins and peptides identified in the STZ and control groups in the PFC and HPC datasets are listed in Appendix A, respectively.

### 2.2. Identification of Differentially Abundant Proteins (DAPs) in the ICV-STZ Model’s Proteome

Protein identification and quantification were performed using the SwissProt database as a reference. Each biological replicate was analyzed separately, and the quantitative comparisons between groups (STZ and control) were conducted, considering at least three biological replicates, as explained in the Materials and Methods section, specifically in the Protein Quantification subsection. Thereby, a total of 1075 proteins identified by at least one unique peptide were quantified in the PFC dataset, encompassing 1063 proteins in the STZ samples and 1060 in control (Figure 2A). In the HPC dataset, 1064 identified by at least one unique peptide were quantified, encompassing 1060 proteins in the STZ samples and 1038 in control (Figure 2B). Similarly to what was found in the qualitative analysis, most proteins were common to both the STZ and control samples in the PFC and in the HPC datasets. The normalized XIC values of each protein is available in Appendix A.

Differential abundance analysis was carried out using only the proteins quantified in both groups (intersections in Figure 2A,B). As a result, significant differential abundances were detected for 116 out of 1048 proteins (11% of blue dots) in the PFC dataset, of which 86 proteins were more abundant, and 30 proteins were less abundant (Figure 2C). In the HPC dataset, significant differential abundances were detected for 81 out of 1034 proteins (7.8% of blue dots), from which 56 proteins were more abundant, and 25 proteins were less abundant (Figure 2D). The fold change of each protein is available in Appendix A. 

A total of 17 proteins were detected as differentially abundant in both PFC and HPC: six proteins were more abundant (Apoe, Aprt, Cst3, Ctsd, Nme1 and Septin8) and five proteins were less abundant (Ckmt1, Cplx2, Nutf2, Phgdh, Rnpep) in both datasets. Rtcb was more abundant in the PFC and less abundant in the HPC dataset and five proteins (Atp5f1a, Atp5f1c, Atp5f1d, Nptxr, and Uqcrc1) were more abundant in the HPC and less abundant in the PFC dataset. Uchl3 exhibited lower abundance in the HPC dataset. However, Uchl3 did not show a statistically significant difference in the PFC dataset (Figure 3).

### 2.3. Identification of Dysregulated Pathways in the ICV-STZ Model

A network representing the functional association and physical interaction of the DAPs was constructed to gain biological insights. As a result, in the PFC, 12 more abundant proteins were classified as part of the ribosome complex, four less abundant proteins as mitochondrial membrane proteins, and four more abundant proteins were identified as related to the chaperone complex (Figure 4A). In the HPC, 14 more abundant proteins were classified as mitochondrial membrane proteins and 5 more abundant proteins as participating in insulin secretion (Figure 4B).

### 2.4. Experimental Validation of mRNAs through qRT-PCR

We also investigated the quantification of the mRNA levels of Uchl3 and Nutf2 through qRT-PCR. The proteins encoded by these mRNAs were differentially abundant in the proteome analysis. The literature has indicated them as related to Aβ plaque formation, and they are implicated as potential targets for therapeutic interventions in neurodegenerative disorders [25,26]. As a result, we identified the up-regulation of both genes in the PFC and the down-regulation of these genes in the HPC. Uchl3 has not shown differential expression in the PFC (*p*-value = 0.34, SHAM-mean = 1.15, STZ-mean = 1.44) and was down-regulated in the HPC (*p*-value = 0.02, SHAM-mean = 1.02, STZ-mean = 0.72) in the mRNA expression analysis, while Nuft2 was down-regulated in both PFC (*p*-value < 0.01, SHAM-mean = 1.19, STZ-mean = 2.44) and HPC (*p*-value < 0.03, SHAM-mean = 1.16, STZ-mean = 0.60), as depicted in Figure 5.

## 3. Discussion

In the present study, we investigated the proteome of the pre-frontal cortex (PFC) and hippocampus (HPC) of the ICV-STZ model to evaluate AD-like molecular hallmarks. Most of the identified proteins were shared between the STZ and control groups in the PFC and HPC datasets. This indicates high similarity between the proteome profile of these two brain regions and that STZ promotes subtle alterations after its administration. These results are consistent with the literature, since several molecular alterations of AD arise only months after STZ administration [16]. Furthermore, differentially abundant proteins (DAP) were identified, with the majority being related to mitochondrial membrane, insulin secretion, ribosome, and the chaperone complex.

Mitochondrial dysfunction has been proposed to precede cognitive decline and promote the aggregation of pathogenic proteins in AD [27]. Multiple studies have implicated the mitochondria as one of the functional links between type 2 diabetes (T2D) and AD [28,29]. Moreover, the ICV-STZ model has been previously reported to produce an AD-like dysregulation of the mitochondria in the PFC and HPC [30,31]. Similar to our findings, Stefanova and colleagues (2019) identified downregulated mitochondrial genes in the PFC of a rat model of LOAD [32], and Reddy and colleagues (2004) identified upregulated mitochondrial genes in the HPC of a transgenic mouse model of AD [33]. Hamezah and colleagues (2018) also identified upregulated proteins of mitochondrial complexes in the HPC of late-aged rats (23 and 27 months) when compared to younger rats (14 months) [34]. Adav and colleagues (2019) identified mitochondrial dysfunction in the prefrontal cortex, specifically in the medial frontal gyrus, of early onset AD patients through quantitative proteomics. They found the downregulation of Sdha, Atp5f1a, Atp5f1c, Atp5f1d, and Uqcrc1 proteins [35]. Interestingly, these proteins were also found to be less abundant in our analysis of the PFC samples. As a mitochondrial dysfunction, insulin resistance is an important hallmark that connects AD and T2D [36], and it also influences AD pathology in non-diabetic AD patients [37]. In our analysis, we identified misregulated proteins involved in insulin secretion. Therefore, we believe that ICV-STZ can be used as a model to further investigate the mitochondrial and insulin secretion roles in LOAD onset in future studies.

Dysregulation of the translation machinery is a risk marker for AD [38]. For instance, Garcia-Esparcia and colleagues (2017) identified upregulated ribosomal genes in the frontal cortex of patients with AD and a rapid course of Alzheimer’s disease (rpAD) patients [39]. Among these genes are Rps10 and Rps13, which were also found to be increased in our proteome analysis of the PFC, along with several other more abundant ribosome proteins. Additionally, Suzuki and colleagues (2022) also reported increased ribosome biosynthesis in the cerebrovascular tissue of AD patients, although the authors did not find significant changes in the neuronal tissue [40]. In addition, we also identified more abundant chaperone proteins in our proteome analysis of the PFC, indicating protein folding dysregulation, as previously observed by Ashraf and colleagues (2014), in the context of AD and T2D [41]. Thus, our results suggest that ribosomes and chaperones might be an important class of proteins to be further in-depth evaluated, and ICV-STZ could be used as a model to help addressing this question.

The Cst3 gene encodes a protein (CysC) highly abundant in brain tissues [42]. This protein is implicated in AD due to its co-localization with amyloid plaques [43]. We detected an increase in this protein in both PFC and the HPC, suggesting a possible involvement in Aβ plaque formation. Nme1 is a protein with a serine/threonine-specific kinase activity that plays a role in neural growth and development [44]. In our analysis, we found this protein more abundant in the PFC and HPC proteome, indicating that it may have an increased activity in early stages of AD, followed by a downregulation in late stages of AD, as demonstrated by Ansoleaga and colleagues (2014) in the entorhinal cortex of AD patients at stages III–IV and V–VI [45]. Additionally, Pedrero-Prieto and colleagues (2019) reported that this protein was exclusively found in AD amyloid-β-enriched extracts when compared to non-AD amyloid-β-enriched extracts [46]. Therefore, Nme1 might also be involved in amyloid-β plaque formation, and its role in the AD onset, as well as its regulation mechanisms in AD, should be further studied.

The reticulons (RTNs) are proteins with a characteristic C-terminal membrane-bound reticulon-homology domain (RHD) [47]. In our analysis, we identified the low abundance of the protein Rtn3 and high abundance of Rtn4 in the PFC. Interestingly, a recent study has demonstrated that Rtn3 deficiency causes increased Bace1 protein levels [48], leading to β-amyloid peptide generation (Aβ) [49]. AD patients also present high levels of Rtn4, suggesting that this protein can serve as a potential biomarker for AD [50]. Calcium/calmodulin-dependent protein kinase II (Camk2) is a multifunctional protein kinase abundant in the central nervous system (CNS), and it becomes active when it binds to Ca^2+^/calmodulin [51]. Similar to our results, Fang and colleagues (2019) identified the downregulation of Camk2a in the hippocampus proteome of AD patients, but its expression level was not significantly altered in the pre-frontal cortex [52]. Another interesting protein in the context of AD is Casein kinase 2, beta polypeptide (Csnk2b). This protein is highly abundant in the neuro fibrillar tangles (NFT) of AD patients [53], being more abundant in our proteome analysis of the PFC.

In the proteome analysis, we identified decreased levels of Nutf2 in the PFC and HPC, and decreased levels of Uchl3 in the HPC. However, Nutf2 was up-regulated in the PFC and down-regulated in the HPC in our mRNA expression analysis through qRT-PCR. Despite Uchl3 not showing a statistically significant change at the transcriptome and proteome levels of the PFC samples, previous reports indicate that the loss of this gene, along with Uchl1, contributes to neurodegeneration [54]. This indicates that Uchl3 translation might be downregulated, decreasing its protein level in the PFC, at least in the ICV-STZ model. The nuclear transport factor 2 (NUTF2), also known as NTF2, plays an important role on the nuclear transportation of proteins [55]. The accumulation of Nutf2 was observed in hippocampal neurons, both with and without tangles, in AD patients, but not in control cases [56]. Moreover, Nutf2 was also found to be increased in the hippocampus of an AD non-human primate model [57]. It has been demonstrated that the nuclear pore complex (NPC) and transport proteins, such as Nutf2, are linked to the formation of NFTs by facilitating the transport of pathological tau proteins from the nucleus to the cytoplasm [26,56]. Although Nuft2 was decreased in both our transcriptome and proteome analysis of the HPC, it was increased in our transcriptome analyses of the PFC. This result suggests that the dysfunction of nucleocytoplasmic transport may manifest differently in the ICV-STZ model compared to AD, and it should be further investigated.

## 4. Materials and Methods

### 4.1. Animals

Male Wistar rats (60–90 days old, weighing 300–350 g), provided by the animal facility of the Federal University of Parana (UFPR), were housed in groups of 3–4 in polypropylene cages with wood shavings as bedding. The rats were maintained at 22 ± 2 °C on a 12 h/12 h light/dark cycle (lights on at 7:00 AM); water and standard chow were available ad libitum. Before the experiment, the rats were allowed to acclimatize for 1 week to reduce environmental stress. The experiments were performed following the Brazilian Law for Animal Experimental Ethics and Care (11.794/8 October 2008) and the guidelines of the UFPR Committee on the care and use of laboratory animals. The experimental procedures were approved by the University Ethics Board (CEUA/BIO-protocol—SCB, UFPR 1411-A).

### 4.2. Stereotaxic Surgery

Stereotaxic surgery was carried out as previously described [17,58,59]. The animals were anesthetized with a 3 mL/kg dose of anesthetic Equitesin (1% sodium thiopental, 4.25% chloral hydrate, 2.13% magnesium sulfate, 42.8% propylene glycol, and 3.7% ethanol in water) and a dose of atropine sulfate (0.4 mg/kg) to reduce the production of secretions). The animals were placed in a stereotaxic apparatus (David Kopf, St Tujunga, CA, USA). A 28-gauge stainless steel needle was lowered into each lateral ventricle (LV). The stereotaxic coordinates for ICV infusion, according to Paxinos and Watson [60], were measured: anterior/posterior, −0.8 mm from bregma; medial/lateral, ±1.5 mm from the midline; dorsal/ventral, −3.8 mm from the skull. An electronic pump (Insight, Ribeirão Preto, SP, Brazil) was used to control the flow of the injections at a rate of 1.0 μL/min over 4.5 min. The lesioned group received bilateral ICV injections of STZ (3 mg/kg total dose) dissolved in sterile 0.9% saline (4.5 μL per injection site). Sham surgery followed the same procedure, but sterile saline was injected instead of STZ. After surgery, all the rats were allowed to recover from anesthesia for 2–4 h in a heated and well-ventilated room. Food and water were placed inside the cage for 10–15 days so that the animals could easily access it without physical trauma caused by the head surgery.

### 4.3. Sample Preparation

The samples were pulverized in liquid nitrogen, as previously described [61]. Protein extraction was then performed using a solution of 0.1% RapiGest (*w*/*v*) in 50 mM triethylammonium bicarbonate (TEAB). Subsequently, the extracted proteins were centrifuged at 18,000× *g*, at 4 °C for 15 min, and the supernatant was collected. The protein content was quantified using a fluorimetric assay on the Qubit 2.0 platform, following the manufacturer’s instructions. Next, 100 µg of total protein from each sample was reduced with 10 mM dithiothreitol (DTT) at 60 °C for 30 min. The samples were then cooled to room temperature and incubated in the dark with 25 mM iodoacetamide (IAA) for 25 min. The samples were subsequently digested overnight with trypsin at a 1:50 enzyme-to-substrate (E/S) ratio, at 37 °C.

### 4.4. Desalting

In due course, the enzymatic reaction was halted by adding trifluoroacetic acid (0.4% *v*/*v* final concentration), and the peptides were incubated for an additional 40 min to degrade the RapiGest. Afterward, the samples were centrifuged at 18,000× *g* for 10 min to remove any insoluble materials. The peptides were then quantified using the fluorometric assay—Qubit 2.0 (Invitrogen, Waltham, MA, USA)—following the manufacturer’s recommendations. Each sample was desalted and concentrated using Stage-Tips (Stop and Go-Extraction Tips), as described by Rappsilber and colleagues [62].

Sample lysis was carried out following the SPEED protocol [63]. Subsequently, sample quantification was performed using the fluorometric Qubit assay, as per the manufacturer’s instructions. One hundred micrograms of each sample were reduced with dithiothreitol (final concentration 10 mM) for 30 min at 60 °C, cooled to room temperature (20 °C), followed by alkylation with iodoacetamide (final concentration 30 mM) for 25 min. Finally, the samples were digested overnight with trypsin in a 1/50 (E/S) ratio, at 37 °C. The reaction was interrupted with trifluoroacetic acid (TFA) 10% (final concentration of 1%), followed by centrifugation for 15 min at 18,000× *g*, and quantified again by the fluorometric Qubit assay. Ten micrograms of each sample were desalted with Stop and Go Extraction Tips (Stage Tips), as described by Rappislber and colleagues (2003) [62].

### 4.5. Mass Spectrometry

Peptides were subjected to LC-MS/MS (liquid chromatography with tandem mass spectrometry) analysis with an UltiMate 3000 (Thermo Fisher^®^, San José, CA, USA) ultra-high-performance liquid chromatography (UHPLC) system coupled with an Orbitrap FusionTM LumosTM mass spectrometer (Thermo, San José), as follows: The peptide mixtures were loaded into a column (75 mm i.d., 30 cm long) packed in-house with a 3μm ReproSil-Pur C18-AQ resin (Dr. Maisch) with a flow of 250 nL/min. Subsequently, they were eluted with a flow of 250 nL/min, from 5% to 40% ACN, in 0.1% formic acid in a 140 min gradient (23). The mass spectrometer was set in data-dependent acquisition (DDA) mode to automatically switch between full-scan (MS) and MS/MS (MS2) acquisition. Survey MS spectra (from m/z 300–1500) were acquired in the Orbitrap analyzer with a resolution of 120,000 at m/z 200. The most intense ions captured in a 2 s cycle time were chosen, excluding unassigned ones that had a 1+ charge state. The ions were sequentially isolated and fragmented using higher-energy collisional dissociation (HCD) with a normalized energy of 30. The fragment ions were analyzed with a resolution of 15,000 at 200 m/z. The general mass spectrometric conditions were as follows: spray voltage, 2.5 kV; no sheath and auxiliary gas flow; ion transfer tube temperature of 250 °C; predictive automatic gain control (AGC) enabled; and S-lens RF level of 40%. Mass spectrometer scan functions and nLC solvent gradients were regulated using the XCalibur 4.1 data system (Thermo, San José). Two technical replicates were acquired for each biological replicate.

### 4.6. Protein Identification

Protein identification was based on the peptide-spectral matching (PSM) approach, using the Comet search algorithm embedded into the freely available PatternLab for Proteomics computational environment (version 5) [64]. Two FASTA files containing non-redundant protein sequences were used in two separate searches: the SpliceProt/SwissProt database (29,044 entries) and SwissProt (9775 entries). The SpliceProt/SwissProt is a customized protein sequence database composed of non-redundant sequences from SpliceProt [65] and SwissProt [66] databases. The first protein search was performed to obtain an overview of the ICV-STZ model’s proteome, and all biological replicates were analyzed together. The second protein search was performed to identify and further compare the quantity of proteins between the two groups: STZ and control (differential abundance analysis). In this case, each biological replicate was analyzed separately.

For all searches, we used a target-reverse database enriched with 128 common MS contaminant sequences (e.g., keratins, albumin, and trypsin). Uninterpreted high-resolution MS/MS spectra were searched against this comprehensive database using Comet default parameters. The enzyme specificity was semi-specific, no proline restriction was specified for trypsin, up to 2 missed cleavages were allowed, and the initial precursor mass tolerance was set to 30 ppm. The following modifications were considered (up to 2 variable modifications per peptide): (1) carbamidomethyl (C, fixed) and (2) oxidation (M, variable). PSM results were filtered by the Search Engine Processor (SEPro) tool implemented in PatternLab for Proteomics. These steps are described in detail in the PatternLab for Proteomics latest bioinformatic protocol [63]. The final post-processing step was adjusted to converge to reliable results showing ≤ 1% FDR at the protein level and ensure that all identifications had less than 10 ppm mass error for precursor tolerance.

### 4.7. Protein Quantification

Protein label-free quantification was performed according to the XIC (extracted ion chromatogram) normalized by TIC (total ion chromatogram) [63] of the identified proteins using the SwissProt database. A minimum of seven MS1 points were accepted for obtaining the extracted ion-chromatogram (XIC). Only proteins that could be quantified by at least two unique peptides were considered in the following analyses: (1) The “Approximately area-proportional Venn Diagrams” module was used to determine which quantified proteins were exclusive to control or STZ samples, and which ones were shared by both classes. (2) The “TFold” module was then used to determine which shared proteins were differentially abundant between control and STZ samples (up- and down-regulated). To be considered exclusive to one class (e.g., STZ), a protein has to be quantified in at least three biological replicates of that class (i.e., STZ) and not should not be present in any of the replicates of the other class (i.e., control). Accordingly, proteins shared by both classes had to be quantified in all three biological replicates of both STZ and control groups to be considered for further statistical analysis of differential abundance. This statistical analysis used the Benjamini–Hochberg (BH) approach with a *q*-value of 0.05 [40]. The tool’s F-stringency parameter was automatically adjusted to 0.10 and the L-stringency value was set to 0.2. F-stringency serves as a fold-change stringency parameter, while L-stringency functions to control the assignment of different abundances for low-abundance proteins.

### 4.8. GO Enrichment and Network Analysis

GO enrichment analysis was performed by comparing all DAPs (test-set) in each dataset (PFC and HPC) to the whole rat genome (reference-set) using software STRING version 11.5 [67]. Proteins were classified into biological processes, molecular functions, and cellular components. GO term frequency differences between test- and reference-sets were statistically assessed by a one-tailed Fisher’s exact test, with the FDR filter set to 5% to correct for multiple testing. The functional and physical association network of the DAPs was also constructed using software STRING with a minimum interaction score of 0.400. The final enrichment results were limited to the most specific terms. For the network construction, all the following active interaction sources were considered: experimentally validated, text mining, databases, co-expression, and neighborhood.

### 4.9. Quantitative Real Time Polymerase Chain Reaction (qRT-PCR)

Total RNA was extracted from the hippocampus and prefrontal cortex using mirVana™ PARIS™ Kit (Life Technologies, Carlsbad, CA, USA). RNA was treated with DNase I—RNAse-free (Thermo Scientific) and reverse-transcribed into cDNA using the High-Capacity cDNA Reverse Transcription Kit (Applied Biosystems, San Francisco, CA, USA), according to the manufacturer’s instructions. Real-time PCR was performed by using triplicates of samples by SYBER green. Levels of gene expression were normalized by the Gapdh gene. The following primers were used Uchl3 (5′ GAGCCCTGAAGAAAGAGCCA 3′ and 5′ TGACCTTCATGTGCACTGGTT 3′), Nutf2 (5′ CATCGTGCCAGCCCCAC 3′ and 5′ GCCTAGTTGGGTTCTGTCGT 3′), Gapdh (5′ GTTACCAGGGCTGCCTTCTC 3′ and 5′ GATGGTGATGGGTTTCCCGT 3′).

## 5. Conclusions

Taken together, this work aimed to describe the proteome of the PFC and HPC using an AD-like non-transgenic animal model (ICV-STZ). Our analysis focused on the initial stages of the AD onset. Proteins associated with mitochondria, insulin secretion, ribosome, and the chaperone complex were identified as DAPs, indicating that these biological pathways are important in the AD onset. Furthermore, mRNAs correspondent to some of these DAPs were investigated in the transcriptome through qRT-PCR. We believe that our results demonstrated that ICV-STZ is a valid model to study the LOAD onset. We also contributed to knowledge regarding the molecular alterations in the first stages of the disease using the ICV-STZ rat model.

## Figures and Tables

**Figure 1 ijms-25-02772-f001:**
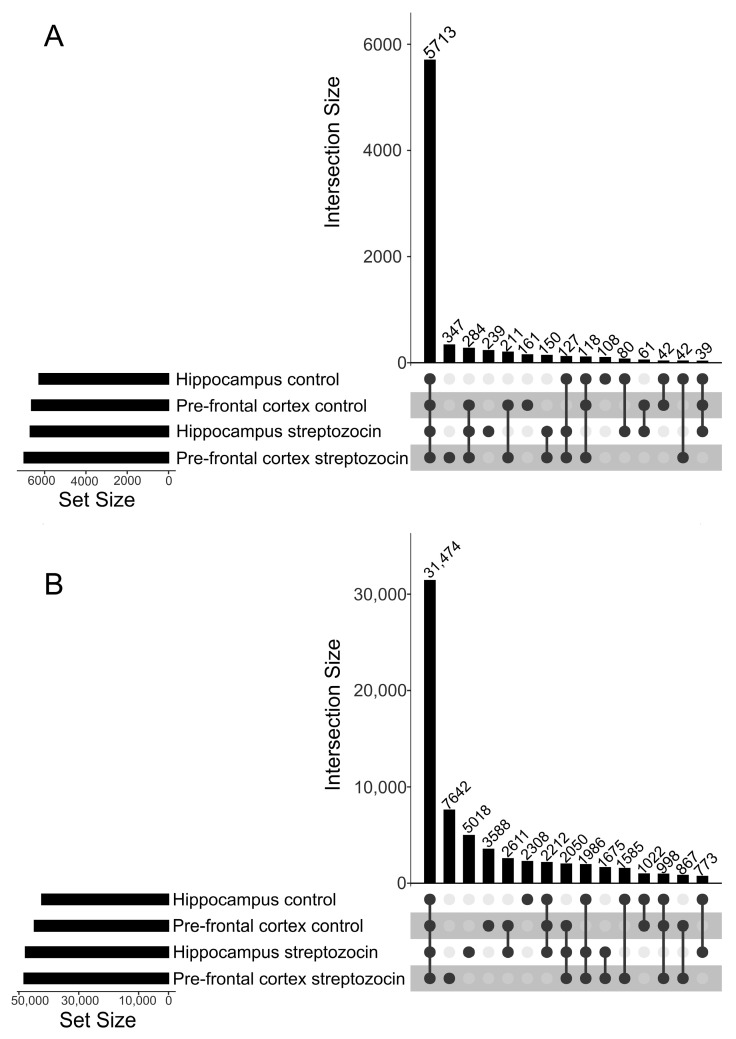
Comparative analysis between PFC and HPC datasets and STZ and control groups. (**A**) Upset plot depicting protein intersections between datasets and groups. (**B**) Upset plot depicting peptide intersections between datasets and groups.

**Figure 2 ijms-25-02772-f002:**
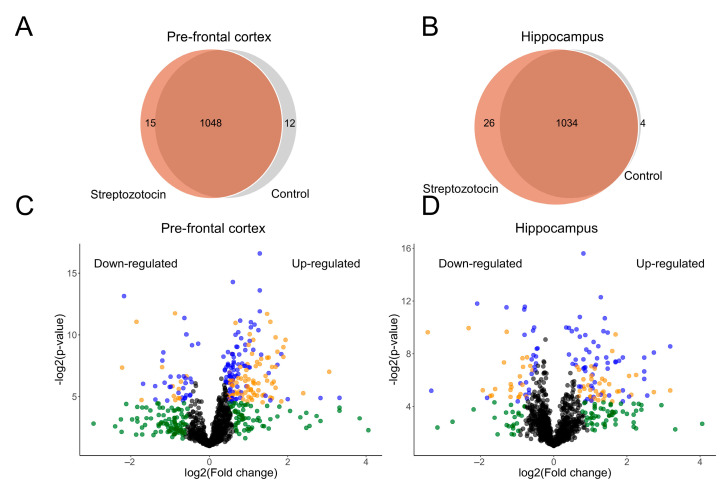
Differentially abundant proteins (DAPs). (**A**) Venn diagram showing the relation between streptozotocin and control samples from the pre-frontal cortex dataset. (**B**) Venn diagram showing the relation between streptozotocin and control samples from the hippocampus dataset. (**C**) Volcano plot showing more/less abundant proteins in the pre-frontal cortex dataset. (**D**) Volcano plot showing more/less abundant proteins in the hippocampus dataset. Blue dots represent proteins that satisfied the fold change, count, and statistical criteria. Orange dots represent proteins that satisfied the fold change and statistical criteria but did not satisfy the count criteria. Green dots represent proteins that only satisfied the fold change criteria. Black dots represent proteins that did not satisfied any criteria. Blue dots with log_2_ fold change above 0 were considered as more abundant proteins, while those below 0 were regarded as less abundant proteins. The F-stringency parameter (fold change criteria) was automatically adjusted to 0.04 for the pre-frontal cortex dataset and 0.09 for the hippocampus dataset. The L-stringency value (count criteria) was set to 0.2 for both datasets and the BH *q*-value (statistical criteria) was set to 0.05.

**Figure 3 ijms-25-02772-f003:**
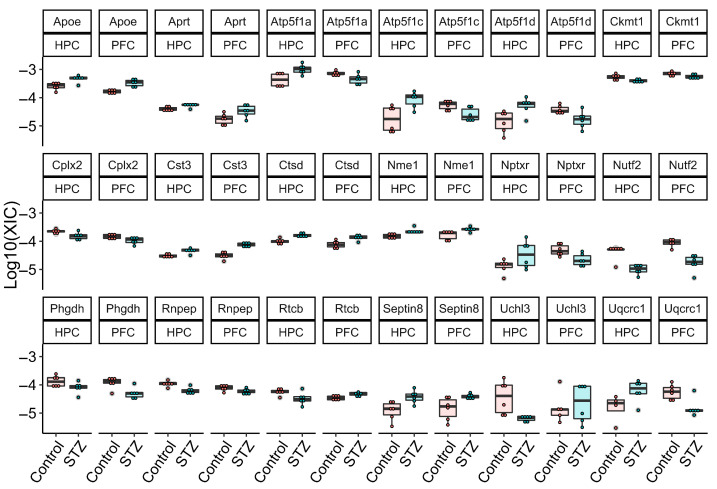
Boxplot depicting the normalized XIC values of each biological sample per DAP in both PFC and HPC datasets. Red dots represent control samples and green dots streptozotocin-treated samples (STZ).

**Figure 4 ijms-25-02772-f004:**
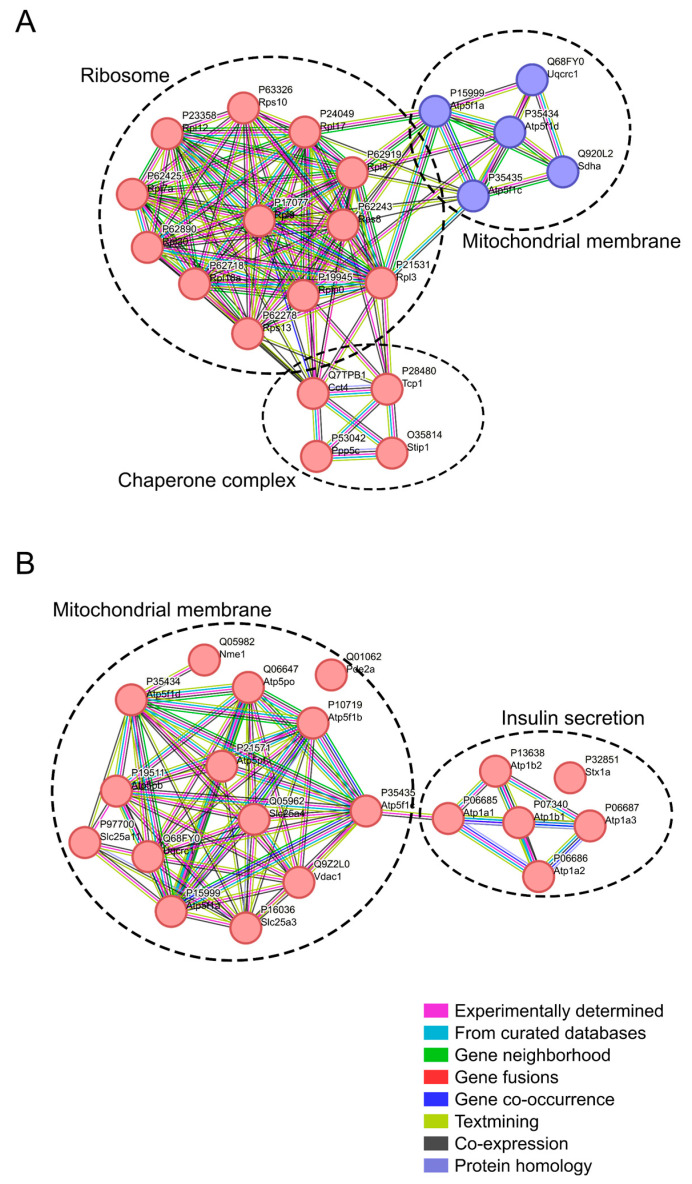
Network analysis of DAPs from PFC and HPC datasets. (**A**) Protein network from PFC dataset. (**B**) Protein network from HPC dataset. Interaction score ≥ 0.400. Red circles represent more abundant proteins and blue circles less abundant proteins. The line colors connecting the proteins represent different evidence types, as depicted in the figure.

**Figure 5 ijms-25-02772-f005:**
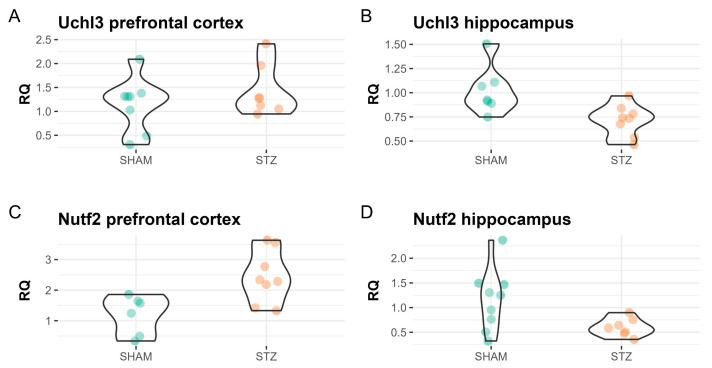
Violin plot of the relative quantification through qRT-PCR. (**A**) Relative quantification (RQ) for the Uchl3 gene in the prefrontal cortex of control samples (SHAM) and treated samples (STZ). (**B**) Relative quantification (RQ) for the Uchl3 gene in the hippocampus of control samples (SHAM) and treated samples (STZ). (**C**) Relative quantification (RQ) for the Nuft2 gene in the prefrontal cortex of control samples (SHAM) and treated samples (STZ). (**D**) Relative quantification (RQ) for the Nutf2 gene in the hippocampus of control samples (SHAM) and treated samples (STZ).

**Table 1 ijms-25-02772-t001:** Quantity of proteins and peptides identified in the proteome datasets.

	Pre-Frontal Cortex		Hippocampus	
	Streptozocin	Control	Streptozocin	Control
N. of total proteins	6992	6629	6693	6269
N. of proteins with unique peptides	2952	2766	2785	2565
N. of peptides	48,798	45,619	48,883	42,977
N. of unique peptides	23,738	21,974	23,494	20,839

## Data Availability

This study’s MS/MS proteomics data are available at the PRIDE (PRoteomics IDEntifications database) repository via the ProteomeXchange Consortium under the accession number PXD048720.

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
