# Peer review of "Proteomic Analysis of a Rat Streptozotocin Model Shows Dysregulated Biological Pathways Implicated in Alzheimer’s Disease"

_ijms, 2024, doi:10.3390/ijms25052772_

Round 1

Reviewer 1 Report

Comments and Suggestions for Authors

The manuscript focused on the proteomic analysis of the Intracerebroventricular (ICV) injection of streptozotocin (STZ) in rats as a model for Alzheimer's Disease (AD). It provided a comprehensive analysis of the proteomic findings in the pre-frontal cortex (PFC) and hippocampus (HPC) of the ICV-STZ model, shedding light on potential molecular hallmarks of Alzheimer's Disease (AD).

The study is well-structured, with a clear progression from the introduction of the ICV-STZ model to the experimental design, results, and discussion of proteomics analysis. The methodology employed is robust. However, there are a few areas that could be improved:

Introduction: While the authors stated why they executed proteomics analysis with the ICV-STZ model, the introduction could benefit from a brief statement of the study's significance or implications for the field to better contextualize the study for readers who may not be specialists in the field.

Methods: In the manuscript, the authors mainly used DDA (Data-Dependent Acquisition) method for proteomics data acquisition and analysis. It would be valuable to know if the authors considered using DIA (Data-Independent Acquisition) method to discover and quantify more proteins for comparison. Additionally, it would be beneficial to explore the consistency of the identified proteins with those reported to be related to Alzheimer’s disease in the literature. If certain proteins were not observed in the list, the use of targeted proteomics could be considered to evaluate these proteins, providing an alternative way to validate the application of the ICV-STZ model in Alzheimer's disease.

Discussion: Addressing the discrepancies between proteomic and transcriptomic data, such as the case of Nutf2 and Uchl3, is important. Discussing the potential reasons for such discrepancies and their implications on AD-related mechanisms could provide valuable insights.

Figure 5: The authors should consider including the XICs (Extracted Ion Chromatograms) and statistical results of Uchl3 and Nutf2 proteins quantification in Figure 5 to validate the reliability of the search results.

Line 120-122: The parameters “F-Stringency” and “L-Stringency” are mentioned without explanation. It would be helpful for readers who are not familiar with these terms to provide an explanation of what they represent and how their values were determined.

Overall, the manuscript presents a valuable contribution to the understanding of AD using the ICV-STZ model. Addressing the points mentioned above can enhance the clarity, significance, and reliability of the findings, making the manuscript even more impactful.

Author Response

Dear Editor,

We are thankful for the anonymous reviewers’ comments and suggestions, which helped to improve the previous version of this manuscript. After reading all of them carefully, we agreed to perform almost all the suggested changes. Please find below our responses to each of the comments.

Response to Reviewer #1 (R1) Comments (C)

The manuscript focused on the proteomic analysis of the Intracerebroventricular (ICV) injection of streptozotocin (STZ) in rats as a model for Alzheimer's Disease (AD). It provided a comprehensive analysis of the proteomic findings in the pre-frontal cortex (PFC) and hippocampus (HPC) of the ICV-STZ model, shedding light on potential molecular hallmarks of Alzheimer's Disease (AD).

The study is well-structured, with a clear progression from the introduction of the ICV-STZ model to the experimental design, results, and discussion of proteomics analysis. The methodology employed is robust. However, there are a few areas that could be improved:

R1-C1. Introduction: While the authors stated why they executed proteomics analysis with the ICV-STZ model, the introduction could benefit from a brief statement of the study's significance or implications for the field to better contextualize the study for readers who may not be specialists in the field.

Response to R1-C1: We appreciate the suggestion. We have added a statement (lines 64-67) as suggested.

R1-C2. Methods: In the manuscript, the authors mainly used DDA (Data-Dependent Acquisition) method for proteomics data acquisition and analysis. It would be valuable to know if the authors considered using DIA (Data-Independent Acquisition) method to discover and quantify more proteins for comparison. Additionally, it would be beneficial to explore the consistency of the identified proteins with those reported to be related to Alzheimer’s disease in the literature. If certain proteins were not observed in the list, the use of targeted proteomics could be considered to evaluate these proteins, providing an alternative way to validate the application of the ICV-STZ model in Alzheimer's disease.

Response to R1-C2: We recognize the benefits of DIA for its enhanced quantitation capabilities and agree that it can potentially provide other identifications. Our preference for DDA is motivated by securing more convincing evidence for peptide identifications as provided by the independent mass spectrum.

Regarding the suggestion to employ targeted proteomics for proteins not observed in our dataset but relevant to Alzheimer's disease, we agree this approach to be of strategical matter for revisiting as well as expanding to future cohorts.

Due to constraints in sample availability and instrumentation time, it is not feasible for us to replicate the data acquisition using DIA or targeted for this manuscript.

R1-C3. Discussion: Addressing the discrepancies between proteomic and transcriptomic data, such as the case of Nutf2 and Uchl3, is important. Discussing the potential reasons for such discrepancies and their implications on AD-related mechanisms could provide valuable insights.

Response to R1-C3: We agree with the reviewer’s comment. To further expand the discussion on this point, we have added a new statement in lines 259-261.

R1-C4. Figure 5: The authors should consider including the XICs (Extracted Ion Chromatograms) and statistical results of Uchl3 and Nutf2 proteins quantification in Figure 5 to validate the reliability of the search results.

Response to R1-C4: We appreciate the suggestion. However, XIC is a quantification method applied to proteomics experiments and the figure 5 depicts the mRNA quantification of the Uchl3 and Nutf2 genes. Therefore, we believe that the XIC values should not be provided on this figure to avoid misleading interpretation of the plot. The Relative Quantification (RQ) of each sample for each gene are represented in the Figure 5 and p-values are provided on the text above the figure 5. Additionally, the normalized XIC values and fold change/p-values of each quantified protein are available at the Supplementary Table S3 and S4, respectively.

R1-C5. Line 120-122: The parameters “F-Stringency” and “L-Stringency” are mentioned without explanation. It would be helpful for readers who are not familiar with these terms to provide an explanation of what they represent and how their values were determined.

Response to R1-C5: Thank you for the suggestion. We added a brief explanation on these parameters (lines 385-387).

R1-C6. Overall, the manuscript presents a valuable contribution to the understanding of AD using the ICV-STZ model. Addressing the points mentioned above can enhance the clarity, significance, and reliability of the findings, making the manuscript even more impactful.

Response to R1-C6: We appreciate the reviewer comment, and we expect to response properly all points raised.

Reviewer 2 Report

Comments and Suggestions for Authors

The manuscript da Silva etal is a proteomics based paper that investigates protein changes in Late-Onset Alzheimer’s Disease models in rats using labelfree quantitation.

Some spelling/grammar corrections:

Line 98: analysis instead of analyzis

Line 120: below instead of bellow

Line 138: representing instead of representing

Line143: mitochondrial instead of mitocondrial

Line 148: connecting instead of conecting

Line149: different instead of different

Line245: Despite Uchl3 not showing a statistically significant change instead of Despite Uchl3 did not show a statistically significant change

It seems that all the up/downregulation of proteins found in this study corresponds well with information from literature. This is great as it confirms that the experimental strategy works, however it lacks novelty.

Why are mRNA levels only experimentally validated for Uchl3 and Nutf2 and not for the other differentially regulated proteins involved in mitochondria membrane, insulin secretion, ribosome and chaperone complex formation?

Nutf2: proteomics data show this protein downregulated in HPC and PFC but in mRNA the protein is upregulated in fig5C for PFC and seems unchanged in HPC in fig 5D

Uchl3: proteomics data show this protein downregulated in HPC and unchanged in PFC which fits the mRNA data

mRNA levels for Nutf2 show the contrary of what the proteomics data detected. This mismatch is addressed in the last paragraph of the discussion but I feel the reasoning needs to be expanded.

This is a typical proteomics paper which reports a list of up and down regulated proteins in their experimental set up. It can be left like this but would gain much more importance if some targets were followed up by other techniques and put into a biological context. However, this might also be beyond the scope and intention of this current manuscript.

The authors have used six biological replicates in each condition which is helpful as it makes the analysis more robust (minimum would be three). They have also controlled their experiment well by minimising variables in their rats, e.g choosing only males and not females. This makes the study valid but at the same time cuts out complementary data such as for female rats.

Comments on the Quality of English Language

The manuscript reads well and is clear. There are minor spelling mistakes.

Author Response

Dear Editor,

We are thankful for the anonymous reviewers’ comments and suggestions, which helped to improve the previous version of this manuscript. After reading all of them carefully, we agreed to perform almost all the suggested changes. Please find below our responses to each of the comments.

Response to Reviewer #2 (R2) Comments (C)

R2-C1. The manuscript da Silva etal is a proteomics based paper that investigates protein changes in Late-Onset Alzheimer’s Disease models in rats using labelfree quantitation.

Some spelling/grammar corrections:

Line 98: analysis instead of analyzis

Line 120: below instead of bellow

Line 138: representing instead of representing

Line143: mitochondrial instead of mitocondrial

Line 148: connecting instead of conecting

Line149: different instead of different

Line245: Despite Uchl3 not showing a statistically significant change instead of Despite Uchl3 did not show a statistically significant change

Response to R2-C1: Thank you for spotting them out. All spelling/grammar corrections were made accordingly.

R2-C2. It seems that all the up/downregulation of proteins found in this study corresponds well with information from literature. This is great as it confirms that the experimental strategy works, however it lacks novelty.

Why are mRNA levels only experimentally validated for Uchl3 and Nutf2 and not for the other differentially regulated proteins involved in mitochondria membrane, insulin secretion, ribosome and chaperone complex formation?

Response R2-C2: We were particularly interested in investigating these two proteins as they are considered as potential targets for therapeutic interventions in neurodegenerative disorders [1–3]. We have added a statement (lines 158-159) to make it clearer.

R2-C3. Nutf2: proteomics data show this protein downregulated in HPC and PFC but in mRNA the protein is upregulated in fig5C for PFC and seems unchanged in HPC in fig 5D

Response to R2-C3: We appreciate the observation. Although the figure 5D seems to be unchanged, the p-valor of this gene is < 0.03 as depicted in the text (lines 164-165). We added the mean of the RQ values to better elucidate the differences between control (SHAM) and STZ groups.

R2-C4. Uchl3: proteomics data show this protein downregulated in HPC and unchanged in PFC which fits the mRNA data

mRNA levels for Nutf2 show the contrary of what the proteomics data detected. This mismatch is addressed in the last paragraph of the discussion but I feel the reasoning needs to be expanded.

Response to R2-C4: We agree with the reviewer’s comment. To further expand the discussion on this point, we have added a new statement in lines 259-261.

R2-C5. This is a typical proteomics paper which reports a list of up and down regulated proteins in their experimental set up. It can be left like this but would gain much more importance if some targets were followed up by other techniques and put into a biological context. However, this might also be beyond the scope and intention of this current manuscript.

Response to R2-C5: We appreciate the comment. Our intention in this study was to offer a preliminary overview of the ICV-STZ proteome based on protein tandem mass spectrometry (MS/MS). As mentioned on the reviewer’s comment, this specific aspect was not within the scope of our manuscript. We hope that future works could benefit from our results to investigate the biological context of ICV-STZ more in depth.

R2-C6. The authors have used six biological replicates in each condition which is helpful as it makes the analysis more robust (minimum would be three). They have also controlled their experiment well by minimising variables in their rats, e.g choosing only males and not females. This makes the study valid but at the same time cuts out complementary data such as for female rats.

Response to R2-C6: We appreciate the comment. Another reason why using male rats is that female rats are less sensitive to streptozotocin toxin, therefore most STZ-induced studies are conducted on male animals [4]. We agree that this is indeed a limiting factor inherent to this model.

REFERENCES

  1. Eftekharzadeh, B.; Daigle, J.G.; Kapinos, L.E.; Coyne, A.; Schiantarelli, J.; Carlomagno, Y.; Cook, C.; Miller, S.J.; Dujardin, S.; Amaral, A.S.; et al. Tau Protein Disrupts Nucleocytoplasmic Transport in Alzheimer’s Disease. Neuron 2018, 99, 925-940.e7, doi:10.1016/j.neuron.2018.07.039.
  2. Gadhave, K.; Bolshette, N.; Ahire, A.; Pardeshi, R.; Thakur, K.; Trandafir, C.; Istrate, A.; Ahmed, S.; Lahkar, M.; Muresanu, D.F.; et al. The Ubiquitin Proteasomal System: A Potential Target for the Management of Alzheimer’s Disease. J Cell Mol Med 2016, 20, 1392–1407, doi:10.1111/jcmm.12817.
  3. Sheffield, L.G.; Miskiewicz, H.B.; Tannenbaum, L.B.; Mirra, S.S. Nuclear Pore Complex Proteins in Alzheimer Disease. J Neuropathol Exp Neurol 2006, 65, 45–54, doi:10.1097/01.jnen.0000195939.40410.08.
  4. Kolb, H. Mouse Models of Insulin Dependent Diabetes: Low‐dose Streptozocin‐induced Diabetes and Nonobese Diabetic (NOD) Mice. Diabetes Metab Rev 1987, 3, 751–778, doi:10.1002/dmr.5610030308.